# DNA Vaccines in the Post-mRNA Era: Engineering, Applications, and Emerging Innovations

**DOI:** 10.3390/ijms26178716

**Published:** 2025-09-07

**Authors:** Praveen Neeli, Dafei Chai, Debanjana Roy, Shivank Prajapati, Srinivasa Reddy Bonam

**Affiliations:** 1Department of Molecular Oncology and Cancer Biology and Evolution Program, H. Lee Moffitt Cancer Center and Research Institute, Tampa, FL 33612, USA; praveen.neeli@moffitt.org; 2Cancer Institute, Xuzhou Medical University, Xuzhou 221004, China; chaidafei@xzhmu.edu.cn; 3Vaccine Immunology Laboratory, Department of Applied Biology, CSIR-Indian Institute of Chemical Technology, Hyderabad 500007, India; roydebanjana15@gmail.com (D.R.); shivankprajapati29@gmail.com (S.P.); 4Department of Microbiology & Immunology, University of Texas Medical Branch, Galveston, TX 77550, USA

**Keywords:** DNA vaccines, mRNA vaccines, cancer, monoclonal antibodies, lipid-nanoparticles, immunogenicity

## Abstract

Deoxyribonucleic acid (DNA) vaccines have re-emerged as a versatile and scalable platform by advances in synthetic biology and delivery systems, positioning them as powerful tools in the post-mRNA vaccine era. Historically considered less potent than viral or mRNA-based platforms, recent breakthroughs have dramatically improved their immunogenicity, safety, and precision. These innovations include synthetic gene circuits, self-amplifying DNA (saDNA), and DNA-encoded monoclonal antibodies (DMAbs), which enable programmable expression and robust immune activation. Clinically, DNA vaccines are expanding into diverse applications, from infectious disease prevention to therapeutic cancer immunotherapy and treatment of immune-mediated conditions. Compared to mRNA vaccines, DNA vaccines offer compelling advantages in terms of thermal stability, ease of manufacturing, and long-term storage. Furthermore, novel adjuvants, electroporation methods, and formulation strategies such as lyophilization and encapsulation continue to broaden their clinical potential. This review explores the full scope of DNA vaccine technology and its engineering foundations, emerging disease applications, and interdisciplinary innovations, while evaluating its comparative performance and future role in global vaccine strategy. With an emphasis on both mechanistic insights and translational feasibility, we propose a roadmap to integrate DNA vaccines into the next generation of precision immunotherapy.

## 1. Introduction

Vaccines remain one of the most transformative interventions in public health, substantially reducing morbidity and mortality from infectious diseases and certain cancers [1]. The rapid development and deployment of messeneger ribonucleic acid (mRNA)-based vaccines during the Coronavirus disease 2019 (COVID-19) pandemic further demonstrated the potential of nucleic acid-based platforms for accelerating vaccine innovation [2]. Despite their clinical success, mRNA vaccines present several inherent challenges, including dependence on ultra-cold storage conditions, relatively low molecular stability, and limited half-life post-administration, all of which restrict their accessibility and scalability in low-resource settings [3]. Deoxyribonucleic acid (DNA) vaccines, first conceptualized in the early 1990s, offer an alternative approach with distinct advantages by delivering plasmid DNA encoding specific antigens directly into host cells [4]. These vaccines harness the body’s machinery to produce target proteins and stimulate both antibody- and T cell–mediated immune responses. Their unique advantages, such as excellent thermal stability, low-cost manufacturing, and ease of large-scale production, make them attractive for use in low-resource settings. While mRNA vaccines have shown significant success in infectious diseases, their application in cancer immunotherapy faces complications related to stability, storage, and tolerance with repeated dosage. mRNA degradation rates vary widely, in human cells, the half-life of mRNA can range from approximately 20 min to several hours, influenced by sequence elements and cellular conditions [5]. Contrarily, DNA vaccines are more easily produced on a large scale, have a higher inherent thermal stability, and are more appropriate for long-term storage and international distribution. Crucially, DNA vaccines can be administered repeatedly since they are typically less reactive than mRNA. This is crucial for cancer patients, who frequently need several boosts to maintain long-lasting anti-tumor immunity.

In addition, DNA vaccines offer broad population coverage through non-HLA-restricted antigen presentation and avoid the complexities associated with viral vector-based vaccines or vaccines that are highly sensitive to cold-chain logistics [6]. Together, these features position DNA vaccines as a flexible, scalable, and globally accessible platform well-suited for addressing both current and emerging health challenges. 

The concept of DNA vaccines traces back to early observations in the 1960s that naked DNA can transfect mammalian cells in vivo, triggering antigen expression [7,8]. Follow-up studies revealed that this could elicit both antibody and cytotoxic T cell responses, establishing the groundwork for DNA-based immunization strategies [9]. Initial clinical trials, such as a 1998 phase I HIV-1 vaccine study, demonstrated immunogenicity but lacked strong efficacy [10]. Subsequent trials for diseases like malaria showed similar promise in generating immune responses in humans [11]. Despite encouraging pre-clinical results, the transition to consistent human efficacy has been hindered by relatively low immunogenicity [12]. While DNA vaccines have seen commercial success in veterinary medicine covering infectious diseases, oncology, and gene therapy, human applications have been slower to advance. Nonetheless, clinical trials have shown progress for targets like Ebola [13], Zika [14], and MERS [15], and the 2021 approval of ZyCoV-D for COVID-19 marked a major milestone [16]. A key logistical benefit of DNA vaccines is their intrinsic thermal stability, as double-stranded DNA is more resistant to degradation compared to single-stranded mRNA. This property reduces reliance on ultra-cold storage, enhancing deployability in low-resource environments, and positioning DNA vaccines as a scalable alternative in global immunization efforts (Table 1).

Recent technological breakthroughs, including nanoplasmid vectors with enhanced expression efficiency and the incorporation of next-generation molecular adjuvants, with advanced delivery methods like electroporation and lipid nanoparticle encapsulation, have dramatically enhanced their immunogenicity and safety profiles, revitalizing interest in DNA-based immunization strategies [19,25,29]. Clinically, DNA vaccines are demonstrating promising efficacy in trials targeting viral infections, cancers, and autoimmune disorders, underscoring their versatility and adaptability [30]. Recently emerging tools, such as self-amplifying DNA constructs, improved synthetic promoters, and co-delivery with novel molecular adjuvants, are poised to further close this gap, expanding the therapeutic scope and efficacy of DNA vaccines in the evolving immunotherapy landscape. This review explores the potential of DNA vaccines in the post-mRNA era, highlighting recent advances like self-amplifying DNA, synthetic gene circuits, and DNA-encoded antibodies. We look at how they are tackling infectious diseases and cancer, and how they stack up against mRNA vaccines. Our goal is to show how DNA vaccines can excel in next-generation immunotherapy, combining science with real-world impact. We searched PubMed, Web of Science, and Scopus up to August 2025, using terms like “DNA vaccines,” “cancer immunotherapy,” and “synthetic circuits.” We picked peer-reviewed studies from 2015 to 2025 that dive into the technological advancements in DNA vaccine engineering, or clinical trials of DNA vaccines, with a few older key studies for context. We also reviewed reference lists of selected articles, preprints and clinical trial registries to capture emerging research. However, only English-language, peer-reviewed publications were included to ensure scientific rigor and reliability.

## 2. Key Technological Advancements

**Electroporation-based DNA delivery**: Efficient delivery of plasmid DNA remains a cornerstone for successful DNA vaccine performance, particularly in achieving robust cellular and humoral immune responses (Figure 1). Among various delivery strategies, electroporation (EP) has emerged as one of the most effective and widely adopted methods, both in preclinical animal models and in human clinical trials [31]. Electroporation enhances DNA uptake by transiently permeabilizing cell membranes through the application of short, controlled electric pulses, thereby facilitating the direct entry of plasmid DNA into the cytoplasm and ultimately into the nucleus for antigen expression [32]. Multiple EP devices have been developed and optimized for human use, adhering to stringent regulatory standards for clinical-grade medical devices. Notably, platforms, such as CELLECTRA^®^ (Inovio Pharmaceuticals), TriGrid™ (Ichor Medical Systems), and DERMA VAX™ (Cyto Pulse Sciences), have been integrated into numerous clinical trials targeting infectious diseases and cancers [33]. These devices vary in electrode configuration, electrical parameters, and application modes (e.g., intramuscular vs. intradermal), but all serve to significantly enhance transgene expression, leading to improved antigen-specific T cells and antibody responses. Optimized electroporation protocols have demonstrated transfection efficiencies up to 98% in primary cells, highlighting the high efficacy of this delivery method [34]. Clinical trials show that device choice affects both immune response and patient comfort. The CELLECTRA^®^ platform, used in phase II/III HPV vaccine trials, generated strong T cell responses but caused mild-to-moderate injection site discomfort [35]. The TriGrid™ system also produced good responses in early HIV and influenza studies but is less widely used now [36,37]. Newer minimally invasive systems, such as electroacupuncture-based electroporation, report better tolerability while still maintaining immune potency [38,39]. Together, these data suggest that clinical device selection should balance efficacy with patient experience. Recent advancements have also introduced alternative EP modalities aimed at minimizing patient discomfort and tissue damage [40]. For example, electroacupuncture-based EP systems utilize minimally invasive, disposable needle arrays to deliver plasmid DNA, offering a more tolerable and potentially scalable option for clinical translation. These approaches have shown promise in animal models and have advanced into early-phase human trials for therapeutic vaccines [41]. In addition to delivery efficiency, EP-based platforms often induce local tissue inflammation and innate immune activation, which act as natural adjuvants to further enhance vaccine immunogenicity [42]. Collectively, these innovations in EP technology represent a critical enabling factor for the success of next-generation DNA vaccines, including those targeting cancer neoantigens, viral pathogens, mutated oncogenic drivers, and others.

**LNP-based DNA delivery**: Despite their efficacy, EP devices pose logistical challenges as they require specialized equipment, trained personnel, and may not be ideal for widespread vaccination efforts, particularly in resource-limited environments (Figure 1). Consequently, there is growing interest in non-electrical DNA delivery systems that offer simpler, scalable, and patient-friendly alternatives [43]. Cationic lipid-based nanoparticles have long been used to facilitate DNA delivery into cells. However, a major recent innovation—spurred by the success of mRNA vaccines—is the use of ionizable lipid nanoparticles (LNPs) for nucleic acid delivery, including DNA [44]. Recent studies have demonstrated that DNA-LNP formulations can generate protective immune responses against SARS-CoV-2, highlighting their potential as next-generation vaccine carriers [45,46,47]. Studies indicate that LNPs can achieve endosomal escape efficiencies ranging from 20% to 80% in vivo, depending on formulation and delivery conditions [48]. However, optimizing LNPs for plasmid DNA presents unique challenges due to its larger size, structural complexity and immunogenicity. Plasmid DNA is larger and more rigid than mRNA which affects its encapsulation efficiency, particle stability and intracellular trafficking [49]. To encapsulate plasmid DNA efficiently, LNPs require higher cationic lipid-to-DNA ratio. High-throughput screening using DNA barcoding has found a number of intriguing formulations that get beyond plasmid DNA’s intrinsic problems, namely its size and strong negative charge. Significant strategies include the use of cationic helper lipids (e.g., DOTAP and DDAB) in conjunction with ionizable lipids (e.g., DLin-MC3-DMA and SM-102) to promote DNA condensation at higher N/P ratios (usually 6:1–12:1), while PEG-lipids and cholesterol help maintain structural stability and regulate particle size. This ensures efficient complexation and compact particle size for effective cellular uptake [50]. Further, researchers have employed DNA barcoding strategies to rapidly screen and refine LNP formulations, improving transfection efficiency and biodistribution [51]. One critical barrier to DNA-LNP delivery is the inflammatory response, often triggered by cyclic GMP-AMP synthase (cGAS)-stimulator of interferon genes (STING) pathway activation due to cytosolic DNA sensing [52]. Recent advancements have demonstrated that co-encapsulation of STING inhibitors or endogenous lipid modulators within LNPs can reduce this inflammation, enabling longer transgene expression and improved safety profiles [53]. Looking ahead, novel LNP formulations specifically engineered for DNA-based cancer vaccines, including those co-delivering immune-stimulatory adjuvants or targeting specific immune cells, are likely to drive a new wave of DNA immunotherapy development. The convergence of rational lipid chemistry, high-throughput screening platforms, and immune modulation strategies positioned the DNA-LNP systems as a transformative tool for both prophylactic and therapeutic vaccine pipelines [54].

**TcBuster transposon-based delivery system:** Along with electroporation and LNP-based delivery system (Non-viral DNA vaccine delivery), Transposon-based delivery can be used to transfer DNA into the host (Figure 2). DNA transposons are naturally occurring DNA transfer vehicles that can be used to integrate DNA into the host genome. DNA transposons move from one genomic location to another with a cut-and-paste mechanism. DNA transposon as a genetic tool can be used to transfer foreign DNA into the host genome [55]. Tularemia, also known as rabbit fever, caused by *Francisella tularensis*, is a zoonotic disease [56]. *Francisella tularensis* LVS (live vaccine strain) was produced using the Tn5-derived transposon [57]. A live attenuated vaccine was generated against diarrheal disease cholera using a transposon [58]. Three major superfamilies of transposons commonly used for gene transfer in human cells are Tc1/mariner, piggyBac (PB), and hAT [55]. TcBuster transposon-based delivery system is a time-saving and cost-effective method to transfer larger cargo when compared to conventional viral vectors [59]. TcBuster tends to integrate more randomly than lentiviral vectors, but it is less biased toward coding exons and transcripts, potentially offering a safer insertional profile. Its integration sites tend to be farther from transcription start site (TSS) reducing the likelihood of activating proto-oncogenes [59]. Genome-wide analyses indicate that TcBuster has a lower preference for integrating near oncogenes compared with other transposon systems, and reported off-target integration rates remain low, supporting its relative safety [59]. Nevertheless, because TcBuster integration is still essentially random, long-term monitoring and careful genomic profiling are recommended to fully mitigate the risk of insertional mutagenesis or unintended proto-oncogene activation.

The system uses two main components, TcBuster mRNA transposase and TcBuster-compatible DNA transposon. TcBuster-M mRNA transposase is an engineered version of the wild-type enzyme delivered as mRNA, or it can be delivered directly as a protein in the cytoplasm along with the TcBuster DNA transposon plasmid that contains the gene of interest flanked by inverted terminal repeats (ITRs). These ITRSs are recognized by the transposase enzyme. Transposases excise the transposon unit and insert it into genomic DNA [60]. Not only vaccines, but several therapeutics have also been developed using this system, including CAR-T, CAR-NK cells, and others. Since transposons integrate into the host genome, they enable stable expression of the target protein, in contrast to mRNA, which provides only transient expression that is typically lost after 2–4 cell divisions [61]. Reported gene integration rates for transposon systems vary; for example, Sleeping Beauty can achieve up to ~10% integration in human cells, while piggyBac shows efficiencies comparable to viral vectors [62,63].

**Self-amplifying DNA vaccines**: Self-amplifying DNA (saDNA) vaccines, also known as DNA replicons, represent a next-generation platform that combines the scalability and safety of plasmid DNA with the potency of RNA replicon-based expression [64]. These vectors encode viral replicase proteins (e.g., non-structural proteins 1–4 from alphaviruses or flaviviruses) and a gene of interest (GoI), enabling cytoplasmic RNA self-replication post-transcription in vivo [65]. This enables higher levels of antigen production from lower doses of DNA, which is especially advantageous in the context of emerging infectious diseases, where rapid and scalable responses are essential, leading to exponentially higher antigen expression compared to conventional DNA vaccines, which often produce up to 10^6^ RNA copies per cell [66].

Unlike traditional DNA vaccines that rely solely on host cell machinery, saDNA introduces a built-in replication mechanism that significantly enhances mRNA stability and protein synthesis, improving immune priming without increasing the plasmid load. Furthermore, platforms, such as the Gemini-D system, demonstrate the feasibility of combining the stability and ease of manufacturing of DNA with the high expression profiles typically associated with RNA vaccines, making saDNA a powerful hybrid tool in the next generation of vaccine technologies [66,67]. Modern saDNA constructs use a eukaryotic promoter (e.g., CMV) to drive expression of a full-length replicon RNA. The replicase complex amplifies the RNA in the cytoplasm, producing large amounts of subgenomic RNA that are translated into antigen. Design optimizations incorporate Foot-and-Mouth Disease Virus (FMDV) 2A cleavage sites to facilitate the separation of the GoI from viral elements, along with the inclusion of self-cleaving ribozymes and synthetic untranslated regions (UTRs) to enhance transcript stability. Additionally, nuclear localization signals (NLS) are employed to promote efficient nuclear import of the DNA [68] 

Preclinical studies confirm that saDNA remains largely episomal with minimal risk of genomic integration [69]. Long-term expression (e.g., >18 months for luciferase in mice), absence of anti-DNA autoantibodies, and no systemic DNA spread have been observed [70]. These features support a strong biosafety profile, particularly for repeat-dose regimens. Although saDNA can sustain antigen expression for extended periods, booster doses may still be required to refresh memory responses and maintain durable protection, especially in cancer immunotherapy. The saDNA vaccines have demonstrated broad efficacy in models of Infectious diseases (Zika, Ebola, SARS-CoV-2, and rabies), several cancers (e.g., HPV, NY-ESO-1, and HER2), and biodefense targets (anthrax toxins) (listed in Table 2). Their ability to elicit potent T cell responses at low doses (100–1000× lower than standard DNA vaccines) makes them highly attractive for both prophylactic and therapeutic use.

**Synthetic DNA Optimization**: Synthetic DNA optimization has become a cornerstone in enhancing the performance of DNA vaccines for infectious diseases and cancer [76,77]. Fine-tuning regulatory elements, such as promoters, introns, and codon usage, has proven instrumental in improving antigen expression and immunogenicity. Promoters like CMV and synthetic alternatives can significantly increase transcription efficiency, especially when paired with enhancer elements [78]. The strategic inclusion of introns, such as CMV intron A, further boosts mRNA processing and stability, leading to higher protein output in host cells [79]. Codon optimization is another vital aspect, which involves reconfiguring gene sequences to match the codon bias of the target organism, thereby enhancing translation accuracy and speed. This has shown marked improvements in responses to antigens like Ag85B in tuberculosis models [80]. Altogether, these optimization strategies collectively enable more potent, durable, and targeted immune responses from DNA vaccine platforms. Advanced codon optimization strategies, such as NeuralCodOpt, leverage deep learning to enhance antigen expression by optimizing translation kinetics and mRNA stability in host cells. Incorporating such AI-driven tools into DNA vaccine design can substantially improve transgene expression, potentially lowering dosage requirements and augmenting immunogenicity [81].

**Synthetic gene circuits**: Synthetic gene circuits in DNA vaccines are engineered genetic constructs designed to regulate the expression of vaccine antigens in a programmable and context-dependent manner (Figure 3) [82]. These circuits are built using the principles of synthetic biology and mimic the logic and control systems found in electronics, like “on/off” switches or AND gates, allowing precise control over gene expression in response to specific biological signals [83,84,85]. In DNA vaccines, synthetic gene circuits enable controlled antigen expression by activating it specifically in target tissues or in response to infection, thereby enhancing safety and reducing off-target effects [86]. They also fine-tune the strength and timing of the immune response, offering precise spatial and temporal regulation [87]. Additionally, these circuits integrate diverse regulatory elements, such as inducible promoters, repressors, and biosensors, to dynamically respond to physiological signals like inflammation, hypoxia, or changes in pH [88]. For example, a DNA vaccine may include a circuit that activates antigen production only in inflamed tissues, or only when co-stimulatory cytokines are present, enhancing both specificity and efficacy. Some circuits have also been proposed to incorporate safety features like self-deleting constructs to prevent prolonged or unintended gene expression [89]. In essence, synthetic gene circuits represent a transformative step toward creating “smart vaccines,” which means the vaccines that can adapt to and interact with the host environment in real time for maximal immunotherapeutic benefit. Recent advances have introduced the use of CRISPR interference (CRISPRi) and synthetic riboswitches in DNA plasmids, allowing real-time gene suppression or activation with tunable response thresholds [90,91]. Furthermore, computationally optimized modular circuit libraries, developed with the help of AI-driven design tools, are now being tested to streamline the assembly of logic-based DNA vaccines for rapid deployment against emerging pathogens [92]. Synthetic RNA-based feedback elements are also being explored as hybrid control systems within DNA vaccines, enabling autonomous adjustment of antigen levels in response to intracellular feedback, minimizing immune overstimulation [93].

**DNA-encoded monoclonal antibodies:** Recent innovations have explored the use of synthetic DNA (synDNA)-encoded monoclonal antibodies, or DNA-encoded monoclonal antibodies (DMAbs), as a transformative strategy to overcome these limitations [94]. This approach delivers plasmid DNA encoding therapeutic antibodies directly into host tissues, typically via electroporation, enabling in vivo antibody production for extended durations (Figure 4) [95]. Preclinical studies have demonstrated that synDNA vectors encoding anti–PD-1 and anti–CTLA-4 antibodies can elicit functional checkpoint blockade with sustained expression and tumor control in mouse models, offering a potentially more accessible immunotherapy delivery method [96,97]. Bispecific antibodies, which bridge tumor antigens and immune effector cells (T cells), are also gaining clinical traction. Blinatumomab, targeting CD19 and CD3, was the first bispecific T cell engager (BiTE) approved by the FDA for relapsed/refractory B-cell acute lymphoblastic leukemia [98]. However, these molecules often face challenges, such as short serum half-lives and high production costs. DNA-encoded delivery of bispecific constructs has emerged as a promising alternative. Notably, in vivo expression of a DNA-encoded bispecific antibody was shown to achieve prolonged expression and effective tumor suppression in preclinical cancer models, suggesting this platform may streamline delivery of complex biologics [99]. The DMAb platform has also shown considerable promise against infectious diseases. Early efforts used plasmid DNA to encode antibody fragments or Fab-like constructs for anti-HIV neutralization [100]. More recently, full-length, broadly neutralizing antibodies against HIV [101,102], influenza A and B [95,103], dengue [104], chikungunya [105], Zika [106], and Ebola [107] viruses have been successfully expressed using DMAbs in mice and non-human primates, demonstrating both functional activity and durable protection. These findings highlight the potential of the platform for rapid, scalable, and cold-chain–free deployment in outbreak settings [108].

In bacterial infections, DMAbs are being investigated as adjuncts or alternatives to traditional antibiotics. For example, a DNA-encoded antibody targeting *Pseudomonas aeruginosa* protected mice from lethal pneumonia caused by multidrug-resistant strains [109]. Similarly, a DMAb targeting the outer surface protein A (OspA) of *Borrelia burgdorferi* successfully blocked Lyme disease transmission from infected ticks in murine models [110]. Collectively, these advancements underscore the versatility and clinical promise of DMAbs as a next-generation biologic delivery strategy, combining the precision of antibody therapies with the logistical simplicity of DNA vaccine platforms. As manufacturing and delivery technologies continue to improve, DMAbs may become a cornerstone for both infectious disease control and immuno-oncology interventions (Table 3).

**Advances in formulation:** Molecular adjuvants have emerged as pivotal tools to precisely modulate immune responses by engaging innate signaling pathways that shape the magnitude and quality of adaptive immunity [116]. By delivering tailored molecular patterns, such as nucleic acids, cytokines, or small-molecule agonists alongside with antigens, these adjuvants enhance antigen presentation and promote durable memory formation while minimizing systemic inflammation [117]. Recent advances have leveraged engineered nucleic acid structures and synthetic ligands that selectively activate pattern recognition receptors like STING, TLRs, and RIG-I, thereby fine-tuning the balance between Th1/Th2 and cytotoxic responses to improve vaccine efficacy against diverse pathogens and cancers [118,119,120]. Moreover, molecular adjuvants are being combined with cutting-edge delivery platforms, such as LNPs and polymeric carriers, to optimize co-localization with antigen-presenting cells, enhancing cellular uptake and controlled release [117,121]. These innovations highlight the potential of molecular adjuvants not only to boost vaccine potency but also to reduce the need for high antigen doses, offering a path toward safer and more effective immunotherapies [117,121].

Emerging technologies in DNA vaccine formulation and stability are transforming the landscape by addressing critical challenges related to storage, delivery, and durability. Techniques, such as lyophilization have been optimized to stabilize DNA vaccines without compromising their structural integrity or immunogenicity, enabling long-term storage at ambient temperatures and easing cold chain requirements [122]. Encapsulation strategies using LNPs, polymeric carriers, and biodegradable hydrogels further protect DNA from enzymatic degradation while enhancing cellular uptake and controlled release at the target site [123]. These innovations not only improve vaccine shelf-life and facilitate global distribution but also enable dose sparing and potent immune activation by ensuring efficient antigen expression. Advances in formulation science thus play a pivotal role in making DNA vaccines more practical and effective, particularly for deployment in resource-limited settings and rapid pandemic response scenarios [124].

**DNA vaccine strategies for various cancers and clinical progress:** Prior to 2019, most clinical investigations involving DNA vaccines for cancer centered on encoding tumor-associated antigens (TAAs), which although found in normal tissues, are typically upregulated or uniquely presented in malignant cells. These TAAs can be grouped into distinct classes, including overexpressed proteins like MUC1, survivin, Ep-CAM, WT1, TAL6, and HER2/Neu; differentiation antigens, such as tyrosinase and prostate-specific antigen (PSA); and cancer/testis antigens like NY-ESO-1 and members of the MAGE family [125]. These molecules have long served as key targets for DNA vaccine platforms aimed at enhancing tumor-specific immune responses. Currently, numerous clinical trials are underway assessing the effectiveness of DNA vaccines that target non-mutated tumor antigens across a variety of cancers. The most extensively studied malignancies in this context include breast, prostate, and cervical cancers (see Table 4). In these studies, vaccines often encode well-characterized antigens, for example, HER2 or Mam-A in breast cancer, PAP in prostate cancer, and HPV-derived E6/E7 oncoproteins in cervical cancer [126]. While DNA vaccines have shown promise in eliciting potent immune activation, especially through the induction of cytotoxic T lymphocytes (CTLs), their standalone efficacy is often limited by the complex immune escape mechanisms employed by tumors. These include the emergence of tumor variants with reduced immunogenicity and the establishment of an immunosuppressive tumor microenvironment (TME) within the TME, populations such as regulatory T cells (Tregs) and myeloid-derived suppressor cells (MDSCs) contribute to T-cell dysfunction and promote immune tolerance. To overcome these barriers and achieve durable anti-tumor effects, DNA vaccines are increasingly being investigated in combination with agents that both amplify antigen-specific immunity and reprogram the immunosuppressive landscape of the TME [127]. We prioritized some of the following cancers due to high disease burden, accessibility of target antigens, and existing immunological characterization. 

**Prostate Cancer:** Prostate cancer remains one of the most actively explored malignancies in DNA vaccine research, with several plasmid-based candidates undergoing clinical evaluation. Among the most studied are pTVG-HP and pTVG-AR, both of which leverage electroporation-enhanced intradermal delivery to maximize antigen uptake and immune activation [128]. The pTVG-HP vaccine encodes prostatic acid phosphatase (PAP), a non-mutated self-antigen expressed in most prostate tumors. Clinical studies have consistently demonstrated that pTVG-HP can safely elicit PAP-specific CD8^+^ T-cell responses, even in patients with advanced or recurrent diseases. A recent randomized phase II trial combining pTVG-HP with GM-CSF reported improved metastasis-free survival in biochemically recurrent prostate cancer (NCT01341652), highlighting its potential as a maintenance immunotherapy [129]. However, when nivolumab (anti–PD-1) was added to the vaccination regimen (NCT03600350), the combination was well tolerated but failed to achieve durable tumor regression, suggesting that immune checkpoint blockade may need further refinement when paired with antigen-specific vaccination in prostate cancer [130].

In parallel, pTVG-AR was developed to target the ligand-binding domain of the androgen receptor (AR), a critical driver of tumor progression, especially in the castration-resistant setting. By generating an immune response against AR-expressing cells, pTVG-AR seeks to complement or extend the therapeutic window of androgen deprivation therapy (ADT). A multicenter phase I study demonstrated that vaccination with pTVG-AR and GM-CSF induced robust Th1-biased immune responses and was associated with prolonged PSA progression-free survival in some patients (NCT02411786). A more recent approach involves combining both pTVG-HP and pTVG-AR with pembrolizumab, an anti-PD-1 antibody, in a multicenter randomized trial (NCT04090528) [131]. This trial is evaluating whether dual-antigen vaccination, coupled with immune checkpoint blockade, can improve progression-free survival (PFS) and delay the onset of metastases in patients with castration-resistant or biochemically recurrent prostate cancer. Interim results suggest an enhanced immune profile and improved antigen breadth, although long-term efficacy data are still pending, with final results anticipated by late 2025. These studies underscore the evolving role of DNA vaccines in prostate cancer, not as monotherapies, but as components of combinatorial immuno-oncology regimens that target tumor antigens while counteracting immune escape mechanisms within the tumor microenvironment [132].

**Breast Cancer:** Breast cancer has emerged as a key focus for DNA vaccine development, particularly for subtypes with poor prognosis or high risk of recurrence. DNA vaccines are being explored not only for their therapeutic potential but also for their role in immune surveillance, reactivation, and recurrence prevention. One of the most innovative approaches in this space is WOKVAC, a multi-epitope DNA vaccine encoding IGFBP2, HER2, and IGF-1R-three proteins implicated in the transition of pre-invasive lesions to aggressive breast malignancies. Delivered via plasmid pUMVC3 and combined with the immunostimulant sargramostim (GM-CSF), WOKVAC has been evaluated in a phase I trial (NCT02780401) involving HER2-negative, node-positive breast cancer patients at high risk of relapse [133]. The study demonstrated that the vaccine was well tolerated and elicited immune responses against all three encoded antigens, suggesting potential for preventing recurrence after standard therapy. Building on these findings, a phase II trial (NCT04329065) assessed WOKVAC in combination with neoadjuvant taxane-based chemotherapy and HER2-targeted monoclonal antibodies (e.g., trastuzumab, pertuzumab) in HER2-positive breast cancer [134]. Administered on day 13 of the first three chemotherapy cycles, WOKVAC was found to increase T-bet^+^ CD4^+^ and CD8^+^ tumor-infiltrating lymphocytes (TILs), key markers of type I immune polarization. Preliminary outcomes indicate the vaccine may improve pathological complete response (pCR) rates and reshape the immune microenvironment, although longer follow-up is needed.

Another promising candidate is STEMVAC, a polyepitope DNA vaccine encoding transcriptional regulators and oncoproteins such as CD105, YB-1, SOX2, CDH3, and MDM2. Initially tested in a phase I trial (NCT02157051) involving patients with advanced-stage, HER2-negative breast cancer, STEMVAC demonstrated safety and immunogenicity, especially in eliciting T-cell responses against non-traditional tumor-associated targets. Unlike earlier antigen-specific vaccines, STEMVAC represents a broader immunological strategy designed to disrupt multiple oncogenic pathways simultaneously.

More recently, phase II trials have expanded STEMVAC’s application beyond advanced disease. For instance, NCT05455658 evaluates the vaccine in patients with early-stage triple-negative breast cancer (TNBC) following standard treatment [135]. The study explores whether post-treatment vaccination can sustain immunological vigilance and reduce recurrence risk. Similarly, another phase II trial (NCT05242965) is testing the vaccine in non-small cell lung cancer (NSCLC) to assess its versatility across epithelial cancers, often in combination with GM-CSF as an adjuvant to boost immune cell recruitment and activation [136]. Together, these trials reflect a shift in breast cancer immunotherapy from passive surveillance to active immunological conditioning, where DNA vaccines are used to amplify tumor-specific T-cell populations, counteract residual disease, and enhance the durability of conventional treatments.

**Cervical Cancer:** Human papillomavirus (HPV) driven cervical cancer continues to represent a major global health burden, particularly in low-resource settings where screening and surgical treatments are limited [137]. DNA vaccines targeting oncogenic viral antigens, specifically HPV E6 and E7 proteins, are at the forefront of therapeutic strategies aimed at preventing the progression of precancerous lesions and treating established disease [135]. Among the most advanced candidates is VGX-3100, a plasmid DNA vaccine developed to treat high-grade squamous intraepithelial lesions (HSILs) associated with HPV-16 and HPV-18, the two most oncogenic HPV genotypes, delivered intramuscularly with electroporation. VGX-3100 has shown the ability to induce robust antigen-specific cytotoxic T-cell responses in multiple early-phase studies [138]. A pivotal phase II trial (NCT01304524) demonstrated significant histopathological regression of cervical intraepithelial neoplasia (CIN 2/3) along with viral clearance, suggesting that therapeutic vaccination could offer an alternative to invasive procedures in selected patients [139].

These findings were further validated in the REVEAL 1 phase III trial, where VGX-3100 achieved meaningful clinical endpoints, including HSIL regression and HPV clearance, when compared to placebo. The vaccine exhibited a favorable safety profile, with most adverse events being mild and limited to the injection site. The ongoing REVEAL 2 trial has completed enrollment, and its top-line results are anticipated to further define VGX-3100′s role as a non-surgical, immunotherapeutic option for cervical HSIL, a notable unmet need in current gynecologic oncology [139].

Another promising therapeutic vaccine in clinical development is GX-188E, which also targets HPV-16/18 E6 and E7 oncoproteins but is designed for use in advanced cervical cancer. In a recent phase II study (NCT03444376), GX-188E was administered in combination with pembrolizumab, a PD-1 checkpoint inhibitor, in patients with persistent or metastatic cervical cancer. The combination yielded an overall response rate (ORR) of 31.3%, including complete responses in 10.4% of patients, and a median overall survival of 16.7 months. Importantly, the regimen was well tolerated, with no unexpected immune-related toxicities. These results have prompted calls for a phase III trial to further evaluate this combination as a potentially transformative approach for patients with limited treatment options [140]. Together, these studies underscore the emerging value of DNA vaccines as both preventive and therapeutic tools in cervical cancer. By harnessing HPV-specific cellular immunity, these platforms may offer a durable, non-invasive strategy to complement or replace surgical management, especially in settings where access to operative care is limited.

**Personalized neoantigen DNA vaccines: precision immunotherapy for tumor-specific targeting:** Traditional TAAs, while widely expressed across cancer types, often share similarity with normal or germline proteins. This overlap can result in immune tolerance, limiting their capacity to activate a strong CTL response. Consequently, many clinical trials employing non-mutated TAAs have yielded modest outcomes, with limited survival benefit when compared to existing standards of care. In contrast, neoantigens arise from somatic mutations unique to an individual’s tumor. These mutations give rise to novel peptides that are not present in healthy tissues, allowing the immune system to recognize them as truly foreign. This lack of central tolerance enables a more robust and tumor-specific immune activation, making neoantigens attractive targets for personalized cancer vaccines [141]. Furthermore, their restricted expression to malignant cells reduces the risk of autoimmune toxicity, a common limitation in TAA-based strategies. The generation of personalized neoantigen vaccines involves integrated genomic and transcriptomic profiling, typically through whole-exome and RNA sequencing of tumor tissue or circulating PBMCs. Once mutations are identified, computational pipelines predict HLA-restricted immunogenic epitopes, which are validated using binding affinity algorithms, peptide-MHC stabilization assays, and immune assays such as ELISpot [142]. These validated epitopes are then encoded into custom-designed polyepitope DNA constructs, allowing for simultaneous targeting of multiple patient-specific mutations in a single vaccine formulation.

Preclinical investigations of such multi-epitope neoantigen DNA vaccines have shown encouraging results across several tumor models [143]. These vaccines have been associated with elevated IFN-γ secretion, strong activation of CD4^+^ Th1 and CD8^+^ CTLs, and suppression of both primary tumor growth and metastatic spread [144]. Importantly, they appear to recondition the tumor microenvironment, converting it into an immunologically active state conducive to T cell infiltration and sustained memory responses. Clinically, several trials are exploring the therapeutic utility of personalized neoantigen DNA vaccines in solid tumors. For instance, NCT02348320 evaluated a vaccine encoding multiple neoepitopes in triple-negative breast cancer, showing immunogenicity and feasibility in a first-in-human setting. Similarly, a trial (NCT03199040) tested this approach in HER2-negative breast cancer patients, although the trial was terminated early. In pancreatic cancer, a notoriously immunosuppressive malignancy, another trial (NCT03122106) investigated a personalized polyepitope DNA vaccine in conjunction with chemotherapy. These studies aim to overcome two central challenges in cancer immunotherapy: intratumoral heterogeneity and the immunological invisibility of TAAs [145,146]. By tailoring the vaccine to the specific mutation profile of each patient, these trials push forward the vision of individualized precision immunotherapy. As sequencing technologies, epitope prediction tools, and DNA delivery systems continue to advance, personalized neoantigen vaccines are poised to become a key pillar in the next generation of tumor-specific immunotherapies, particularly when combined with immune checkpoint blockade, cytokine modulation, or adoptive T cell therapy (All clinical trials listed in Table 4).

## 3. Conclusions and Future Directions

This paper presents a comparative approach with mRNA vaccines, highlighting particular opportunities and problems, in addition to consolidating recent developments in DNA vaccine engineering and clinical use. By combining translational methods with mechanistic insights, it advances our knowledge of how DNA vaccines can fit into the next wave of precision medicine and worldwide immunization.

To establish DNA vaccines as next-generation immunotherapeutics, it is essential to move beyond conventional frameworks. Existing platforms can be quickly adapted to target new or emerging infectious diseases. Future efforts should integrate systems biology, precision immunology, and translational science to optimize antigen design, delivery strategies, and immune profiling.

High-throughput technologies can guide the rational development of vaccines tailored to specific disease contexts and patient populations. Precision immunology will enable personalized approaches by aligning vaccine components with individual immune landscapes.

Equally important is bridging technical innovation with clinical feasibility. Scalable manufacturing, improved delivery platforms, and predictive biomarkers will be critical for clinical success. With continued innovation and translational focus, DNA vaccines have the potential to become a versatile platform for both prophylactic and therapeutic applications across infectious, autoimmune, and malignant diseases. 

## Figures and Tables

**Figure 1 ijms-26-08716-f001:**
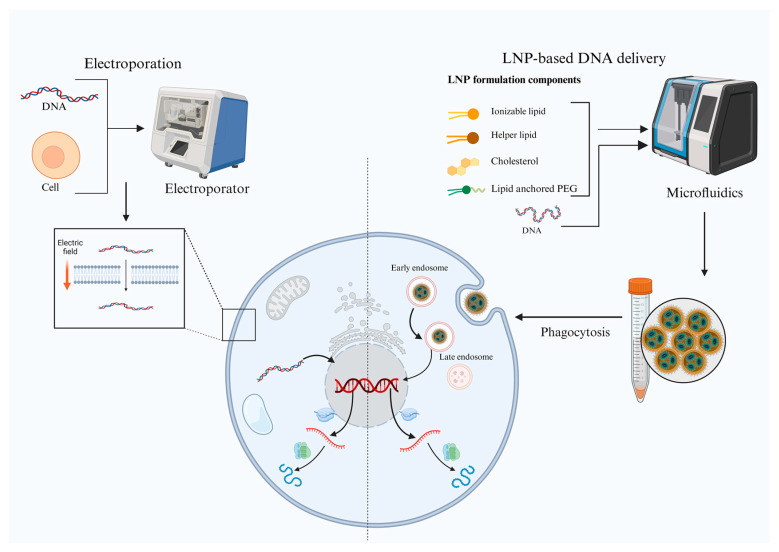
Diagrammatic representation contrasting two methods of delivering DNA vaccines. Lipid nanoparticle (LNP)-based DNA delivery (right) and electroporation-based DNA delivery (left). An electric field momentarily permeabilizes the cell membrane during electroporation, enabling DNA uptake. On the other hand, LNPs made of PEG, cholesterol, helper lipids, and cationic/ionizable lipids are created using microfluidics and taken up by cells through phagocytosis. After entry, endosomes release DNA into the cytoplasm, where it is subsequently expressed. Both strategies aim to enhance intracellular delivery and antigen expression.

**Figure 2 ijms-26-08716-f002:**
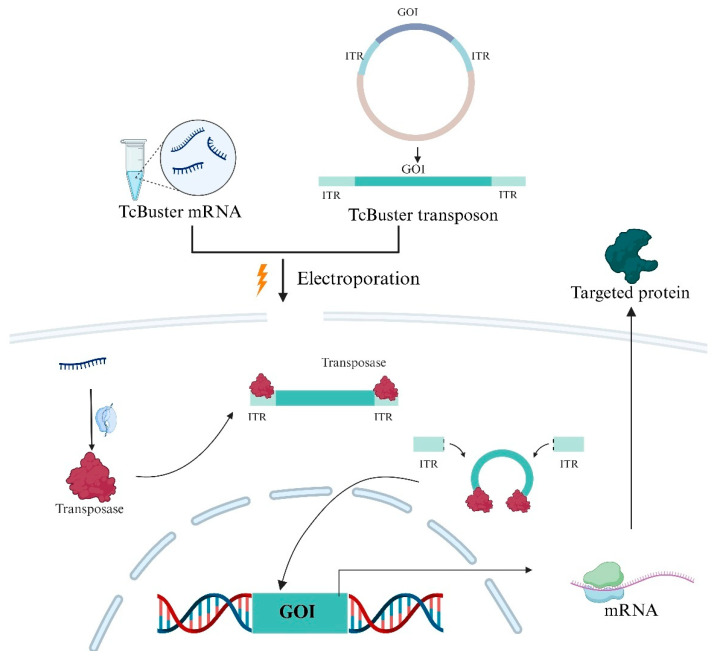
Diagrammatic illustration of electroporation-based TcBuster transposon-based gene delivery. TcBuster mRNA is co-delivered into cells together with a transposon plasmid that contains the gene of interest (GOI) flanked by inverted terminal repeats (ITRs). The GOI is integrated into the host genome through the translation of mRNA into transposase, which occurs inside the cell. This permits the targeted protein to be produced and mRNA to express itself steadily. Effective and sustained transgenic expression is made possible by the system.

**Figure 3 ijms-26-08716-f003:**
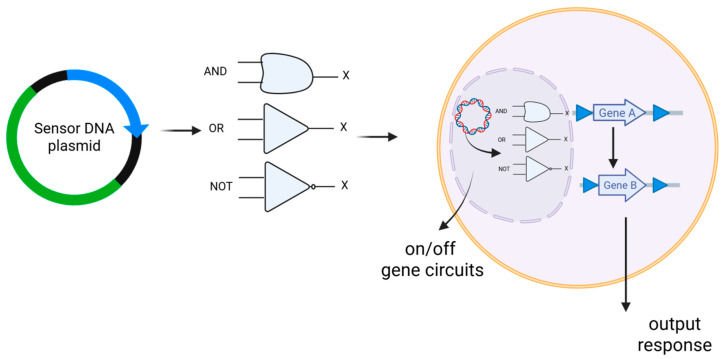
Schematic illustration of DNA vaccines functioning as synthetic gene circuits within host cells to enable precise cancer immunotherapy. The plasmid enters the cell, where logic gates (AND, OR, NOT) process inputs and control therapeutic gene expression. This enables precise, programmable activation of immune responses against cancer.

**Figure 4 ijms-26-08716-f004:**
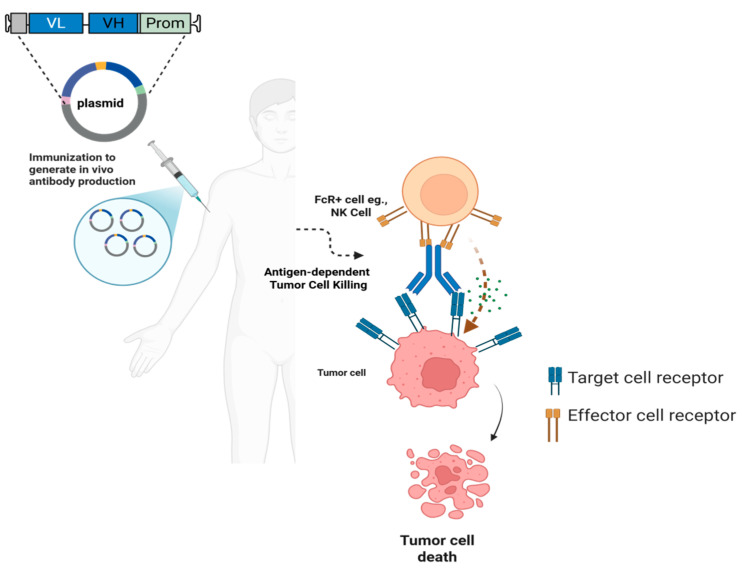
Figure depicting the in vivo DNA-encoded monoclonal antibody (mAb) production: The plasmids designed for immunization, generate VL, VH, sequences that assemble into functional antibodies in vivo. The antibodies can be detected by FcR+ cells (e.g., NK cells), triggering destruction of the targeted tumor cells via antibody-dependent cellular cytotoxicity (ADCC).

**Table 1 ijms-26-08716-t001:** Comparison of Deoxyribonucleic acid (DNA) and Messenger ribonucleic aid (mRNA) vaccines for cancer therapy.

Feature	DNA Vaccines	mRNA Vaccines
**Stability**	Stable at 2–8 °C; can be lyophilized for easier transport and longer shelf life [17]	Requires ultracold storage (−20 °C to −70 °C), complicating logistics [3]
**Delivery target**	Requires entry into the nucleus for transcription [6]	Requires only cytoplasmic delivery for translation [18]
**Delivery systems**	Typically delivered via electroporation, gene gun, or viral vectors [19]	Often delivered in lipid nanoparticles (LNPs) for enhanced stability and cell uptake [20]
**Production**	Relatively inexpensive; scalable with bacterial fermentation systems [19]	Fast production, initially higher cost, but increasingly optimized for large-scale manufacturing [21]
**Mechanism of action**	Transcribed into mRNA in the nucleus, then translated into an antigenic protein in the cytoplasm [22]	Direct translation of mRNA into protein in the cytoplasm [21]
**Immune response**	Induces both cellular and humoral responses; often Th1-biased [23]	Strong inducer of both humoral and cellular immunity, particularly CD8^+^ T cell responses [24]
**Adjuvant requirement**	Often requires co-delivery with adjuvants to boost immunogenicity [25]	May not need separate adjuvants due to innate immunostimulatory properties of RNA and LNPs [26]
**Safety profile**	Very low risk of genomic integration, especially with improved non-integrating plasmid vectors [27]	No integration risk: mRNA is transient and degraded by normal cellular processes [28]

**Table 2 ijms-26-08716-t002:** Self-amplifying DNA (replicon DNA) vaccine studies.

Disease/Target	Vector Format	Dose (μg of DNA)	Key Findings	References
Tumor Neoantigens + PSMA	SFV replicon DNA plasmid	20	Induced tumor regressions and long-term tumor-free survival in murine models	[71]
HPV E6/E7	SFV replicon DNA plasmid	25	Strong CTL responses and tumor control in HPV-positive mouse tumor model	[71]
HIV Env, Gag-Pol-Nef	SFV replicon DNA plasmid	50	Robust antibody and CTL responses, enhanced by protein boost	[72,73]
SARS-CoV-2 Spike	DREP-S (SFV-based replicon DNA)	10	High IgG and neutralizing antibody titers, strong T cell immunity in mice	[74]
Influenza HA	CMV/SFV replicon DNA plasmid	5	Enhanced immune responses vs. conventional DNA vaccine at lower doses	[64,75]

Abbreviations: CMV—Cytomegalovirus, CTL—Cytotoxic T lymphocyte, HA—Hemagglutinin, HIV—Human immunodeficiency virus, HPV—Human Papillomavirus, PSMA—Prostate-specific membrane antigen, SFV—Semliki Forest virus, and SARS-CoV-2—severe acute respiratory syndrome coronavirus 2.

**Table 3 ijms-26-08716-t003:** Selected preclinical and clinical studies of DNA-encoded biologics (recent to old).

Target	Disease Etiology	Biologic Class	Key Findings	References
SARS-CoV-2 virus	Infectious	DMAb	Phase 1 trial in humans shows durable (72-week) expression, stable levels, and tolerable safety profile	[111]
SARS-CoV-2 virus	Infectious	DNA-encoded bispecific	BNT142 RNA-encoded T-cell engager shows preclinical efficacy	[112]
IL-13Rα2 (glioblastoma)	Malignancy	DNA-encoded bispecific T-cell engager (dBTE)	In vivo DNA-launched bispecific engage T cells, control heterogeneous GBM	[113]
ufgHER2	Malignancy	Her2/GP96 vaccine	Anti-HER2 DMAb alone, or bispecific HER2/CD3, controls ovarian tumors and extends survival in mice	[114]
Chikungunya virus	Infectious	DMAb	Single injections protect mice; combo DNA + DMAb offers rapid + durable protection	[105]
*Pseudomonas aeruginosa*	Infectious	DMAb	DMAb protects from lethal pneumonia and works synergistically with antibiotics	[109]
Zika virus	Infectious	DMAb	*In vivo* plasmid delivery generates neutralizing ZIKV antibodies; protects mice from lethal challenge	[106]
*Borrelia burgdorferi*	Infectious	DMAb	OspA-targeting DMAb blocks tick-borne Lyme transmission in mice	[115]
HIV-1	Infectious	Broadly neutralizing Abs	Multiple bNAbs expressed in mice/NHPs at functional levels	[100,102]
Influenza A/B	Infectious	DMAb	Single-dose DMAbs protect mice from lethal influenza challenge	[95,103]
Ebola virus	Infectious	DMAb	DNA-encoded EBOV mAbs confer full protection in mouse models	[107]
Dengue virus	Infectious	DMAb	Multivalent DMAb delivery neutralizes all DENV serotypes; blocks ADE	[104]
PD-1	Malignancy	DMAb	Anti-PD-1 DMAb expressed rapidly, sustained in serum; enhances checkpoint blockade in mice	[96]
CTLA-4	Malignancy	DMAb	DNA-encoded anti-CTLA-4 induces tumor shrinkage in mouse models	[97]
HIV-1	Infectious	F(ab)	DNA-encoded VRC01-like F(ab) yields rapid in vivo expression post-electroporation	[100]
HIV-1	Infectious	Ig-like molecule	Proof-of-concept study for DNA-based delivery of anti-HIV immunoadhesins and in vivo modulation of protein function	[101]

Abbreviations: ADE refers to Antibody-Dependent Enhancement; bNAbs are Broadly Neutralizing Antibodies; CTLA 4 stands for Cytotoxic T-Lymphocyte Antigen 4; dBTE denotes DNA-encoded Bispecific T-cell Engager; DENV is Dengue Virus; DMAb refers to DNA-encoded Monoclonal Antibody; EBOV represents Ebola Virus; F(ab) is Fragment Antigen-Binding (an antibody fragment); GBM stands for Glioblastoma Multiforme; GP96 is Glycoprotein 96; HER2 refers to Human Epidermal Growth Factor Receptor 2; HIV-1 denotes Human Immunodeficiency Virus Type 1; Ig-like molecule stands for Immunoglobulin-like Molecule; IL 13Rα2 refers to Interleukin-13 Receptor Alpha 2; OspA is Outer Surface Protein A of *Borrelia burgdorferi*; PD 1 represents Programmed Cell Death Protein 1; SARS-CoV-2 is Severe Acute Respiratory Syndrome Coronavirus 2; and ZIKV stands for Zika Virus.

**Table 4 ijms-26-08716-t004:** Comprehensive DNA vaccine clinical trials for cancer.

Sponsor/Collaborator	Vaccine (Brand)	Encoded Antigens	Delivery + Combination	Cancer Type and Phase (NCT)	Start	Status/Key Results
Dana-Farber/Yale	Personalized DNA neoantigen vaccine	Patient-specific neoantigens	IM + EP; monotherapy	Advanced kidney, phase I (NCT Pending)	2025	Remission in 9/9 patients ≥3 yrs
Washington Univ (TNBC)	Personalized polyepitope DNA vaccine	4–20 patient neoantigens	IM + EP; monotherapy	TNBC, phase I (NCT02348320)	2024	87.5% RFS at 36 mo
Nature ’24 RCC study	Neoantigen-targeting DNA vaccine	RCC driver mutations	IM + EP; monotherapy	Renal cell carcinoma, phase I	2024	Strong T-cell responses
WUSTL	Personalized neoantigen	Neoantigens	IM EP	Pediatric brain tumor, phase I (NCT03988283)	2024	Not yet recruiting
Wash U (GBM)	INO-5410	LAMP1, IE-1, pp65, gB	IM + EP	Glioblastoma, phase I (NCT05698199)	2023	Recruiting
Wash U (GBM)	Personalized neoantigen DNA w/retifanlimab	Patient-specific neoantigens	IM + EP; + retifanlimab	Unmethylated GBM, phase I (NCT05743595)	2023	Recruiting
University of Washington	STEMVAC	CD105, YB-1, SOX2, CDH3, MDM2	/	Lung NSCLC, phase II (NCT05242965)	2023	Recruiting
Immunomic Therapeutics	ITI-1001	IE-1, pp65, gB	LAMP1, IM EP	Glioblastoma, phase I (NCT05698199)	2023	Recruiting
WUSTL	Personalized neoantigen	Neoantigens	Retifanlimab, IM EP	Glioblastoma, phase I (NCT05743595)	2023	Recruiting
Wash U (SCLC)	Personalized neoantigen DNA + durvalumab	Patient-specific neoantigens	IM + EP; + durvalumab	Small-cell lung cancer, phase II (NCT04397003)	2022	Recruiting
University of Washington	WOKVAC	IGFBP2, HER2, IGF1R	Paclitaxel, Trastuzumab, Pertuzumab, ID	Breast cancer, phase II (NCT04329065)	2022	Recruiting
University of Washington	STEMVAC	CD105, YB-1, SOX2, CDH3, MDM2	rhu GM-CSF, ID	Breast cancer, phase II (NCT05455658)	2022	Recruiting
WUSTL	Personalized neoantigen	Neoantigens	Durvalumab, IM EP	SCLC, phase II (NCT04397003)	2022	Recruiting
University of Wisconsin, Madison	pTGV-AR	AR LBD	Degarelix, Nivolumab, ID	Prostate cancer, phase I/II (NCT04989946)	2021	Recruiting
NCI	pNGVL4a CRTE6E7L2	HPV E6/E7/L2	EP	HPV-16 positive cervical neoplasia, phase I (NCT04131413)	2020	Recruiting
WUSTL	Personalized neoantigen	Neoantigens	INO-9012, IM EP	Glioblastoma, phase I (NCT04015700)	2020	Active, not recruiting
Geneos Therapeutics	GNOS-PV02	Personalized neoantigen	INO-9012, Pembrolizumab, ID EP	HCC, phase I/II (NCT04251117)	2020	Active, not recruiting
University of Wisconsin, Madison	pTGV-HP + pTGV-AR	PAP + AR LBD	Pembrolizumab (a-PD1 Ab), ID	Castration-resistant prostate cancer, phase II (NCT04090528)	2019	Recruiting
INOVIO	VGX-3100	HPV E6/E7	/	Cervical HSIL, phase III (NCT03721978)	2019	Completed
WUSTL	Neoantigens	Neoantigens	Durvalumab, IM EP	TNBC, phase I (NCT03199040)	2019	Terminated
WUSTL	Neoantigens	Neoantigens	Darvalumab, Tremelimumab, IM EP	Renal cell carcinoma, phase II (NCT03598816)	2019	Withdrawn
INOVIO	INO-5401 + INO-9012	WT1, PSMA, hTERT	IM + EP; + cemiplimab, RT	Glioblastoma, phase I/II (NCT03491683)	2018	OS 18–32 mo; immune active
University of Wisconsin, Madison	pTGV-HP	PAP	Nivolumab (a-PD1 Ab), GM-CSF, ID	Prostate cancer, phase II (NCT03600350)	2018	Active
INOVIO	INO-5410	WT1, PSMA, hTERT	Cemiplimab, radiation, chemo, INO-9012, IM EP	Glioblastoma, phase I/II (NCT03491683)	2018	Active
INOVIO	INO-9012	WT1, PSMA, hTERT	INO-9012, Atezolizumab, IM EP	Urothelial carcinoma, phase I/II (NCT03502785)	2018	Active
INOVIO	VGX-3100	HPV E6/E7	/	Anal neoplasm, phase II (NCT03499795)	2018	Completed
INOVIO	VGX-3100	HPV E6/E7	/	CIN 2/3, phase II (NCT01304524)	2018	Completed
NCI	MEDI0457 (INO-3112)	HPV E6/E7	INO-9012, Durvalumab, IM EP	HPV-16/18 cancers, phase II (NCT03439085)	2018	Active, not recruiting
Genexine	GX-188E	HPV E6/E7	Pembrolizumab, IM EP	Cervical cancer, phase I/II (NCT03444376)	2018	Completed
BMS/Bavarian Nordic	Neoantigens	Neoantigens	Nivolumab, Ipilimumab, Prostvac, IM EP	Metastatic prostate cancer, phase I (NCT03532217)	2018	Completed
NCI	pING vector	Neoantigens + mesothelin	Chemotherapy, IM EP	Pancreatic cancer, phase I (NCT03122106)	2018	Terminated
University of Wisconsin	VGX-3100	HPV E6/E7	IM + EP; monotherapy	Cervical/Anal (NCT03185013, etc.)	2017	Completed
University of Washington	WOKVAC: pUMVC3-IGFBP2-HER2-IGF1R	IGFBP2, HER2, IGF1R	Carboplatin, Paclitaxel, ID	Ovarian cancer, phase II (NCT03029611)	2017	Terminated
INOVIO	VGX-3100	HPV E6/E7	/	Cervical cancer, phase III (NCT03185013)	2017	Completed
Genexine	GX-188E	HPV E6/E7	GX-I7, Imiquimod, IM	Cervical cancer, phase/(NCT03206138)	2017	Unknown
University of Washington	WOKVAC	IGFBP2, HER2, IGF1R	rhu GM-CSF, ID	Breast cancer, phase I (NCT02780401)	2016	Active
Washington Univ (Breast)	Mam-A DNA vaccine	Mammaglobin-A	IM + EP	Breast cancer, phase I (NCT02204098)	2015	Recruiting
University of Wisconsin, Madison	pTGV-AR	Androgen Receptor LBD	GM-CSF, ID	Prostate cancer, phase I (NCT02411786)	2015	Completed
University of Washington	STEMVAC	CD105, YB-1, SOX2, CDH3, MDM2	/	Breast cancer, phase I (NCT02157051)	2015	Active, not recruiting
Genexine	GX-188E	HPV E6/E7	/	Cervical cancer, phase II (NCT02596243)	2015	Unknown
WUSTL	Personalized polyepitopes	/	IM EP	TNBC, phase I (NCT02348320)	2015	Completed
University of Wisconsin, Madison	pTGV-HP	Prostatic acid phosphatase (PAP)	rhGM-CSF, ID	Prostate cancer, phase II (NCT01341652)	2011	Completed

Abbreviations: a-PD1 Ab—Anti-Programmed Death-1 Antibody, CDH3—Cadherin 3, CIN 2/3—Cervical Intraepithelial Neoplasia Grades 2/3, CTLA 4—Cytotoxic T-Lymphocyte Antigen 4, DMAb—DNA-encoded Monoclonal Antibody, DENV—Dengue Virus, EBOV—Ebola Virus, EP—Electroporation, F(ab)—Fragment Antigen-Binding (antibody fragment), GBM—Glioblastoma Multiforme, GM-CSF—Granulocyte-Macrophage Colony-Stimulating Factor, GX-188E—Genexine’s DNA Vaccine Targeting HPV Antigens, gB—Glycoprotein B (CMV Antigen), HER2—Human Epidermal Growth Factor Receptor 2, HCC—Hepatocellular Carcinoma, HIV-1—Human Immunodeficiency Virus Type 1, HPV—Human Papillomavirus, HSIL—High-Grade Squamous Intraepithelial Lesion, ID—Intradermal, IE-1—Immediate Early 1 (CMV Antigen), IGF1R—Insulin-like Growth Factor 1 Receptor, IGFBP2—Insulin-like Growth Factor Binding Protein 2, IL 13Rα2—Interleukin-13 Receptor Alpha 2, IM—Intramuscular, Ig-like molecule—Immunoglobulin-like Molecule, INO-9012—INOVIO’s DNA-based IL-12 Adjuvant, LAMP1—Lysosome-Associated Membrane Protein 1, LBD—Ligand Binding Domain, Mam-A—Mammaglobin-A (Breast Cancer Antigen), MDM2—Mouse Double Minute 2 Homolog, NSCLC—Non-Small Cell Lung Cancer, OspA—Outer Surface Protein A (of *Borrelia burgdorferi*), OS—Overall Survival, PAP—Prostatic Acid Phosphatase, PD 1—Programmed Cell Death Protein 1, pING—Plasmid Immunogenic Neoantigen Gene Vaccine Vector, pNGVL4a CRTE6E7L2—Plasmid DNA Vaccine Encoding HPV Antigens, Prostvac—PSA-targeting Poxvirus-based Vaccine, PSMA—Prostate-Specific Membrane Antigen, RCC—Renal Cell Carcinoma, rhGM-CSF—Recombinant Human Granulocyte-Macrophage Colony-Stimulating Factor, rhu—Recombinant Human, rhu GM-CSF—Recombinant Human Granulocyte-Macrophage Colony-Stimulating Factor, RFS—Relapse-Free Survival, SCLC—Small Cell Lung Cancer, SARS-CoV-2—Severe Acute Respiratory Syndrome Coronavirus 2, SOX2—SRY-Box Transcription Factor 2, STEMVAC—Stem Cell Antigen Vaccine (Targeting Embryonic/Stem Markers), TNBC—Triple-Negative Breast Cancer, VGX-3100—INOVIO’s HPV DNA Vaccine Targeting E6/E7, WOKVAC—Washington Ovarian Cancer Vaccine, WT1—Wilms’ Tumor 1 (Tumor Antigen), YB-1—Y-box Binding Protein 1, ZIKV—Zika Virus.

## Data Availability

All data supporting the findings of this study are available in the cited literature.

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
