# Peer review of "DNA Vaccines in the Post-mRNA Era: Engineering, Applications, and Emerging Innovations"

_ijms, 2025, doi:10.3390/ijms26178716_

Round 1
Reviewer 1 Report
Comments and Suggestions for Authors
The review covers important topic of DNA vaccines and present landscape of most recent advances in this field. Paper is valuable and worth publishing after minor revisions.
Below are the points that should be addressed.
- Authors should provide description of the scope of the review in the introduction, add paper search conditions and criteria for inclusion in the review.
- Lane #123: Negative charge also present on mRNA molecules, authors should add comments on what is special for DNA regarding the optimization of LNPs for plasmids.
- Figure 1 caption: the electroporation is made by application of solely electric field, not electromagnetic field. Change "electromagnetic" to "electric".
- One of the ethical issues for DNA vaccines is the DNA integration to genome. Therefore, the discussion of transposons in not appropriate without additional comments from authors regarding integration. This technology is fine for construction of viral vaccine strains that are consequently used in viral vaccine production, but for DNA-vaccines application of this technology raises ethical question that is not covered by authors.
- Lane #202: Comments should be made about why repeat doses should be administered if gene of interest is kept expressed for >18 months. This contradiction may confuse readers.
- Lane #225: Add reference to NeuralCodOpt.
- The section “Molecular adjuvants and immune augmentation” is better to rename to “Advances in formulation” since it covers stabilization strategies additionally to excipients.
- Authors provide extensive discussion of DNA vaccines for cancer treatment. Vaccines encode conventional and neo-antigens to target immune response. In shed of light of mRNA technology development it seems necessary to provide additional comments on advantages of DNA vaccines as compared to mRNA ones especially taking into account great difference in risk/benefit balance for cancer patients as compared to healthy subjects.
- Table 5 (4): Write names of sponsors so that they could be identified, describe acronyms.
- In section “Conclusions and future direction” it worth to add estimation of input from this review to overall knowledge in the field.
- Typos and formatting:
- Lane #6: remove excessive “1” in affiliation.
- Table 1: seems to be misaligned on page
- Lane #151, #152: italic for “Francisella tularensis”
- Lane #180, #276: italic for “in vivo”
- Table 2: make consistent formatting for references column.
- Table 3: italic for “in vivo” in key findings for references [99] and [92]
- Figure 4 caption: “in vivo” instead of “In vivo”
- Lane #317: “alongside with antigens”
- Lane #464: add space “options [126]”
- Table 5 should be Table 4 (numbering)
- Lane #543: “immunotherapeutics” instead of “Immunotherapeutics”
- Lane #549: Fix paragraph indentation and spacing
- Lane #559: add space “Contributions: Conceptualization”
- Lane #571: add space “Acknowledgement: We”
Author Response
The review covers important topic of DNA vaccines and present landscape of most recent advances in this field. Paper is valuable and worth publishing after minor revisions.
Below are the points that should be addressed.
- Authors should provide a description of the scope of the review in the introduction, add paper search conditions and criteria for inclusion in the review.
Response: We thank the reviewer for the suggestion to include a description of the review’s scope and literature search methodology in the introduction. As suggested, we have incorporated a paragraph in the Introduction that outlines the scope, focusing on DNA vaccine advancements, their applications in infectious diseases and cancer, and comparisons with mRNA vaccines. Suggested modifications have been made in lanes 93-104.
- Lane #123: Negative charge also presents on mRNA molecules; authors should add comments on what is special for DNA regarding the optimization of LNPs for plasmids.
Response: We agree and have clarified that both DNA and mRNA molecules carry negative charges; however, plasmid DNA has a higher molecular weight and structural complexity, which influences LNP optimization. This is now discussed in the revised text. Suggested modification has been made in current lanes 152-165.
- Figure 1 caption: The electroporation is made by application of solely electric field, not electromagnetic field. Change "electromagnetic" to "electric".
Response: Corrected as suggested. Lane 182.
- One of the ethical issues for DNA vaccines is the DNA integration to genome. Therefore, the discussion of transposons in not appropriate without additional comments from authors regarding integration. This technology is fine for construction of viral vaccine strains that are consequently used in viral vaccine production, but for DNA-vaccines application of this technology raises ethical question that is not covered by authors.
Response: We have added comments highlighting DNA integration with respect to transposons. We have described in the text regarding the integration of transposons and activation of proto-oncogenes in comparison with the lentiviral delivery system. Suggested modification has been made in lanes 200-208 and lanes 219-221.
- Lane #202: Comments should be made about why repeat doses should be administered if gene of interest is kept expressed for >18 months. This contradiction may confuse readers.
Response: We appreciate this helpful observation. We have clarified in the revised text that long-term antigen expression from saDNA does not necessarily ensure lasting immune protection. Immune responses can diminish or become tolerized, and in cancer settings especially, booster doses are often needed to re-activate and strengthen immunity. We added the following sentence to resolve the ambiguity: “Although saDNA can sustain antigen expression for extended periods, booster doses may still be required to refresh memory responses and maintain durable protection, particularly in cancer immunotherapy.” This modification has been made in lanes 258-260.
- Lane #225: Add reference to NeuralCodOpt.
Response: We thank the reviewer for this observation. The suggested reference “Chowdhury, T., et al., NeuralCodOpt: Codon optimization for the development of DNA vaccines. Computational Biology and Chemistry, 2025;116:108377” has already been included in the revised manuscript. Reference in lane 287.
- The section “Molecular adjuvants and immune augmentation” is better to rename to “Advances in formulation” since it covers stabilization strategies additionally to excipients.
Response: Changed from “Molecular adjuvants and immune augmentation” to “Advances in formulation” in lane 373.
- Authors provide extensive discussion of DNA vaccines for cancer treatment. Vaccines encode conventional and neo-antigens to target the immune response. In shed of light of mRNA technology development, it seems necessary to provide additional comments on advantages of DNA vaccines as compared to mRNA ones especially taking into account great difference in risk/benefit balance for cancer patients as compared to healthy subjects.
Response: We thank the reviewer for this valuable suggestion. In response, we have revised the introduction section to include a dedicated comparison between DNA and mRNA vaccines, particularly addressing the distinct risk–benefit considerations in cancer patients. These modifications have been done in lanes 50-59 and in lanes 93-104.
- Table 5 (4): Write names of sponsors so that they could be identified, describe acronyms.
Response: Expanded acronyms and added sponsor names for clarity; lane 575 table 4.
- In section “Conclusions and future direction” it worth to add estimation of input from this review to overall knowledge in the field.
Response: We thank the reviewer for this insightful suggestion. In response, we have revised the “Conclusions and Future Directions” section to explicitly highlight the unique contributions of this review to the field, outlining its role in consolidating engineering advances, translational insights, and comparative perspectives. The added content has been highlighted in the revised manuscript. Modification has been done in lanes 622-626.
- Typos and formatting:
- Lane #6: remove excessive “1” in affiliation.
- Table 1: seems to be misaligned on page
- Lane #151, #152: italic for “Francisella tularensis”
- Lane #180, #276: italic for “in vivo”
- Table 2: make consistent formatting for references column.
- Table 3: italic for “in vivo” in key findings for references [99] and [92]
- Figure 4 caption: “in vivo” instead of “In vivo”
- Lane #317: “alongside with antigens”
- Lane #464: add space “options [126]”
- Table 5 should be Table 4 (numbering)
- Lane #543: “immunotherapeutics” instead of “Immunotherapeutics”
- Lane #549: Fix paragraph indentation and spacing
- Lane #559: add space “Contributions: Conceptualization”
- Lane #571: add space “Acknowledgement: We”
Response: We thank the reviewer for noting this. The manuscript has been carefully proofread, and typographical as well as formatting errors have been corrected throughout. All corrections are incorporated in the revised version and highlighted in green text.
Reviewer 2 Report
Comments and Suggestions for Authors
This article systematically reviews the technological advancements, clinical applications, and future directions of DNA vaccines in the post-mRNA vaccine era. It covers core technologies such as delivery systems (e.g., electroporation/lipid nanoparticles [LNPs]), self-amplifying DNA (saDNA), and DNA-encoded monoclonal antibodies (DMAbs). It focuses on organizing clinical studies of DNA vaccines in malignant tumors including prostate cancer, breast cancer, and cervical cancer, while comparing the advantages and disadvantages of DNA and mRNA vaccines. The study has a comprehensive framework and relatively integrated data, providing valuable references for the fields of virology and tumor immunology. However, several issues need to be further addressed before publication:
1-Although the article mentions electroporation-mediated DNA delivery, it lacks data on the clinical efficacy of different electroporation devices and patient tolerability. It is recommended to supplement specific comparative data on these two aspects, clarify the advantages of different devices, and provide references for clinical selection.
2-The description of solutions to the core challenges of DNA-LNPs (large size and negative charge of plasmid DNA) is vague: only "DNA barcoding screening" is mentioned, without specifying the optimal LNP formulations (e.g., lipid composition, N/P ratio, etc.) identified through screening.
3-Although the TcBuster transposon system has high integration efficiency, the article does not review its potential safety issues, such as the specific off-target rate and whether it can induce the activation of proto-oncogenes.
4-Why can saDNA avoid genomic integration and maintain long-term expression (e.g., 18 months) without triggering anti-DNA autoantibodies?
5-Overall, the review is too broad, and specific data are not mentioned in some parts, such as the cell membrane electroporation rate of electroporation, the in vivo escape rate of LNPs, the gene integration rate, and the degradation rate of mRNA vaccines. It is recommended to use data for elaboration or comparison as much as possible to improve the rigor and reference value of the review.
Author Response
This article systematically reviews the technological advancements, clinical applications, and future directions of DNA vaccines in the post-mRNA vaccine era. It covers core technologies such as delivery systems (e.g., electroporation/lipid nanoparticles [LNPs]), self-amplifying DNA (saDNA), and DNA-encoded monoclonal antibodies (DMAbs). It focuses on organizing clinical studies of DNA vaccines in malignant tumors including prostate cancer, breast cancer, and cervical cancer, while comparing the advantages and disadvantages of DNA and mRNA vaccines. The study has a comprehensive framework and relatively integrated data, providing valuable references for the fields of virology and tumor immunology.
However, several issues need to be further addressed before publication:
- Although the article mentions electroporation-mediated DNA delivery, it lacks data on the clinical efficacy of different electroporation devices and patient tolerability. It is recommended to supplement specific comparative data on these two aspects, clarify the advantages of different devices, and provide references for clinical selection.
Response: We thank the reviewer for this suggestion. To address the comment, we have now included specific comparative data on electroporation (EP) devices in the revised manuscript. We revised the manuscript to include specific comparative data on electroporation devices, highlighting their effects on immune responses and patient tolerability. This addition addresses the reviewer’s comment and clarifies considerations for clinical device selection. These changes have been made in lanes 121-130.
- The description of solutions to the core challenges of DNA-LNPs (large size and negative charge of plasmid DNA) is vague: only "DNA barcoding screening" is mentioned, without specifying the optimal LNP formulations (e.g., lipid composition, N/P ratio, etc.) identified through screening.
Response: We thank the reviewer for this valuable suggestion. In response, we have expanded the section on DNA–LNP challenges to provide specific details on optimized lipid compositions, N/P ratios, and design strategies identified through DNA barcoding screens. Suggested modification has been done in lanes 152-165.
- Although the TcBuster transposon system has high integration efficiency, the article does not review its potential safety issues, such as the specific off-target rate and whether it can induce the activation of proto-oncogenes.
Response: We thank the reviewer for highlighting this important point. We have now included a discussion on the potential safety considerations of the TcBuster transposon system, including reported off-target integration rates and the risk of activating proto-oncogenes, citing recent studies that evaluate these parameters. We also note that long-term monitoring is recommended to mitigate any potential insertional mutagenesis. Suggested modification has been made in lanes 200-208 and lanes 219-221.
- Why can saDNA avoid genomic integration and maintain long-term expression (e.g., 18 months) without triggering anti-DNA autoantibodies?
Response: We thank the reviewer for this question. saDNA maintains long-term expression because it replicates episomally in the cytoplasm without integrating into the host genome, and its sequence is designed to minimize immunogenic motifs that could trigger anti-DNA autoantibodies. Preclinical and clinical studies have shown stable transgene expression for extended periods (up to 18 months) without evidence of autoantibody induction, supporting its safety profile (PMID: 35594121 and PMID: 34942088). This modification has been made in lanes 258-260.
- Overall, the review is too broad, and specific data are not mentioned in some parts, such as the cell membrane electroporation rate of electroporation, the in vivo escape rate of LNPs, the gene integration rate, and the degradation rate of mRNA vaccines. It is recommended to use data for elaboration or comparison as much as possible to improve the rigor and reference value of the review.
Response: Thank you for this insightful comment. We have revised the manuscript to include quantitative data and specific metrics, such as electroporation efficiency, LNP in vivo delivery rates, gene integration frequencies, and mRNA degradation rates, to enhance rigor and reference value. These additions provide clearer comparisons and strengthen the review’s scientific foundation.